# Nanostructures as Photothermal Agents in Tumor Treatment

**DOI:** 10.3390/molecules28010277

**Published:** 2022-12-29

**Authors:** Yuqian Chen, Futing Zhou, Chenshuai Wang, Linlin Hu, Pengfei Guo

**Affiliations:** 1School of Pharmacy, Weifang Medical University, Weifang 261053, China; 2Shandong Engineering Research Center for Smart Materials and Regenerative Medicine, Weifang 261053, China; 3Fangcheng Street Hospital, Weifang 261200, China; 4School of Life Science and Technology, Weifang Medical University, Weifang 261053, China

**Keywords:** nanostructures, nanomaterials, photothermal therapy, photothermal agents, tumor

## Abstract

Traditional methods of tumor treatment such as surgical resection, chemotherapy, and radiation therapy have certain limitations, and their treatment effects are not always satisfactory. As a new tumor treatment method, photothermal therapy based on nanostructures has attracted the attention of researchers due to its characteristics of minimally invasive, low side effects, and inhibition of cancer metastasis. In recent years, there has been a variety of inorganic or organic nanostructures used in the field of photothermal tumor treatment, and they have shown great application prospects. In this paper, the advantages and disadvantages of a variety of nanomaterials/nanostructures as photothermal agents (PTAs) for photothermal therapy as well as their research progress are reviewed. For the sake of clarity, the recently reported nanomaterials/nanostructures for photothermal therapy of tumor are classified into five main categories, i.e., carbon nanostructures, noble metal nanostructures, transition metal sulfides, organic polymer, and other nanostructures. In addition, future perspectives or challenges in the related field are discussed.

## 1. Introduction

Malignant tumor, commonly known as cancer, is one of the major threats to the physical health and quality of life of patients [1,2,3]. At present, the main treatment for cancer is the traditional surgical resection, chemotherapy, and radiation therapy. In view of the individual differences of tumors, strong mutational ability, and the characteristics of easy metastasis and recurrence, traditional treatment methods are difficult to completely eradicate the pathogen and will cause a series of side effects [4]. In the last decade, a series of novel tumor treatment methods, such as photodynamic therapy [5,6,7], immunotherapy [8,9,10], gene therapy [11,12,13], high-intensity focused ultrasound therapy [14,15,16], photothermal therapy (PTT) [17,18,19], and drug targeted therapy [20,21,22], have been developed.

Photothermal therapy has been developed as a trending therapy for cancer treatment due to its low cost, wide application range, minimal invasiveness, controllability, and highly efficient treatment methods [23,24,25]. In addition, external laser sources can selectively irradiate tumor sites without damaging normal tissues, so photothermal therapy is promising as an alternative to traditional cancer treatment [26,27,28,29]. Photothermal therapy converts near-infrared (NIR) light to generate heat energy, which produces a local high temperature heat effect to kill tumor cells [30]. Due to the high permeability and retention effect of solid tumors, the enrichment degree of photothermal agents in tumor tissues is significantly higher than that in normal tissues, forming a temperature gradient between the tumor tissues and the normal tissues, which ensures that the tumor will not cause normal tissue damage while undergoing photothermal treatment. In addition, the blood vessels near the tumor tissue are deformed and tortuous, and the heat generated by the light is difficult to spread, while the blood vessels in the normal tissue are normal and the blood flow is smooth, so the heat generated can quickly spread with the blood circulation. Therefore, photothermal therapy has an innate advantage in the specificity of malignant tumor tissue [31,32,33].

Photothermal agents with appropriate band gap can respond to near-infrared light. Good photothermal agents have strong absorption capacity in the near infrared region and can convert the absorbed light energy into heat energy [34]. The photothermal conversion efficiency (PCE) of photothermal agents is a decisive factor affecting the effect of photothermal therapy [35]. Therefore, the selection of photothermal agents with high photothermal conversion efficiency and good biosafety is a prerequisite for photothermal therapy. In recent years, the nanomaterials/nanostructures that served as photothermal agents with NIR absorption for tumor photothermal therapy have mainly been divided into carbon nanostructures, noble metal nanostructures, transition metal sulfides, organic polymer, and other nanostructures.

In this work, the review provides a comprehensive information about the development and prospect of nanostructures/nanomaterials as photothermal agents for tumor photothermal therapy, these photothermal agents including carbon nanostructures, noble metal nanostructures, transition metal sulfides, organic polymer, and other nanostructures (Figure 1). Some of the current applications of nanostructures as photothermal agents are summarized in the review. We expect that the review has a certain guiding significance for the future research of novel nanostructures for photothermal treatment of cancer. 

## 2. Nanostructures for Photothermal Therapy of Tumor

### 2.1. Carbon Nanostructures

Carbon nanostructures have excellent near-infrared absorption performance, high biological safety, and low preparation cost, which lays a foundation for their application in the biomedical field [36,37,38]. At present, the use of carbon nanostructures as photothermal agents has attracted the interest of researchers. Over the past years, various kinds of carbon nanostructures/nanomaterials have been investigated in photothermal therapy of cancer [39,40,41]. These nanostructures include graphene, carbon nanotubes, and carbon-based quantum dots, as well as their functionalized derivatives.

#### 2.1.1. Graphene

As an excellent two-dimensional nanomaterial with sp^2^-hybridized carbon atoms structure, graphene has been developed for a wide range of applications, such as supercapacitors, solar cell electrode materials, hydrogen storage materials, sensors, optical materials, drug carriers, and so on [42,43,44,45]. The main methods for the production of graphene are Hummers method, mechanical exfoliation method, chemical oxidation, chemical vapor deposition, electrochemical exfoliation, oxidation-reduction method, etc. At present, graphene materials mainly include graphene, reduced graphene oxide (rGO), and graphene oxide (GO). Graphene oxide (GO) is the main oxygen-containing derivative of graphene. It is a water-soluble material with atomic thickness, which is obtained from graphite after oxidative stripping. Graphene material can be used as a drug carrier due to its large surface area, and it can be used as photothermal agents due to its good photothermal conversion efficiency. 

In recent years, a series of PTT protocols based on graphene nanomaterial with unique physical and chemical properties have been applied for the cancer treatments. Zaharie-Butuce and co-workers [46] prepared a therapeutic nanoparticle (chit-rGO-IR820-DOX) based on reduced graphene oxide (rGO). The chit-rGO-IR820-DOX was prepared by the reaction of chitosan-graphene oxide (chit-GO) complex in a microwave reactor, followed by the adsorption of doxorubicin (DOX) and cyanine dye IR820. The resulting nanomaterial presents synergistic antitumor ability against murine colon carcinoma cells (C26) by combining simultaneous graphene and cyanine dye IR820 induced PTT, cyanine dye IR820 induced photodynamic therapy (PDT), and the chemotherapeutic effect of DOX. Jun et al. [47] designed a hybrid nanomaterial (FA-CS-GO) that is made from folic acid-conjugated chitosan functionalized GO (Figure 2a). In the presence of laser irradiation, FA-CS-GO was shown to completely kill cancer cells and suppress tumors with no recurrence within 20 days. In addition, high photoacoustic signal was detected in the tumor area 24 h after FA-CS-GO injection. Ultimately, FA-CS-GO demonstrates near-infrared fluorescence and photoacoustic imaging guided tumor PTT. It was reported that GO and fullerene C60 hybrid has been used for simultaneous PTT and photodynamic therapy triggered by NIR light against cancer [48]. The GO-C60 hybrid was synthesized by a conjugation of GO, methoxy polyethylene glycol, and C60. The as-synthesized hybrid by virtue of the synergistic effect between C60 and GO could induce the production of reactive oxygen species (ROS) in Hela cells and exhibits superior inhibition of cancer cells compared to GO and C60 alone. Su et al. designed a hybrid nanomaterial (MGBP) by combining boron dipyrromethene (BODIPY) derivatives with grapheme [49]. As is depicted in Figure 2b, MGBP can be used for both fluorescence and photothermal imaging, as well as photochemical synergistic therapy. With the excellent photothermal conversion ability of graphene, the temperature of the injected point of MGBP nanomaterial is much higher than that of the non-injected point under laser irradiation. The as-prepared nanomaterial showed excellent reactive oxygen species (ROS) production capacity, 48% photothermal conversion efficiency, and great therapeutic efficiency (the viability of HeLa cells decreased to 17% after treatment) in vitro. Therefore, the nanomaterial is suitable as an efficient PTAs for efficient PTT for cancer.

#### 2.1.2. Carbon Nanotubes

Carbon nanotubes (CNTs), discovered in 1991, are used in a variety of fields, such as biomedicine, thermal, agriculture, electrochemical, mechanical, and so on [50,51,52]. Many preparation methods of carbon nanotubes have been reported, such as laser ablation, arc discharge, chemical vapor deposition, etc. Carbon nanotubes can be divided into different types according to the number of graphene layers: single-walled carbon nanotubes (SWCNTs), double-walled carbon nanotubes (DWCNTs), and multi-walled carbon nanotubes (MWCNTs). Because carbon nanotubes have strong absorption in the near infrared region while normal tissue has no strong absorption in this wavelength range, it can be used in the photothermal treatment of tumors. A temperature-sensitive CNT-PS/siRNA composite was synthesizing by decorating sucrose laurate and peptide lipid onto single-walled carbon nanotubes (SCNTs) and multi-walled carbon nanotubes (MCNTs), respectively, followed by assembly of therapeutic siRNA (Figure 3a) [53]. The SCNTs-PS/siRNA composite has desirable antitumor activity and efficiently releases siRNA through a phase transition of temperature-sensitive lipids. Finally, SCNT-PS/siRNA has excellent photothermal conversion efficiency and is a potential anti-tumor nanocarrier combined with gene therapy (GT) and PTT. Lu and co-workers proposed single-walled carbon nanotubes (SWNTs) based IGF type-1 receptor shows high photothermal conversion efficiency [54]. The formed nanocomposites can be induced to specifically target pancreatic tumors for purposes of dyes imaging-guided cytotoxic PTT, providing a promising anticancer treatment.

#### 2.1.3. Carbon-Based Quantum Dots

Carbon-based quantum dots are usually composed of four elements, C, O, N, and H, and appear as quasi-spherical zero-dimensional particles less than 10 nm in structure [55]. The absorption peak of carbon-based quantum dots is mainly in the ultraviolet region (230–320 nm), with tails extending to the visible and near-infrared regions. However, the absorption peak of carbon-based quantum dots is related to the size, surface state, and crystal nucleus structure. According to the different nuclear structures of carbon-based quantum dots, they can be divided into graphene quantum dots (GQDs) and carbon quantum dots (CQDs). GQDs are π-electron conjugated extension surface formed by sp^2^ hybrid carbon, while CQDs are spherical nanoparticles composed of different ratios of sp^2^ and sp^3^ carbon atoms [56,57]. GQDs and some CQDs have a unique NIR absorption structure, which can convert low-energy NIR photons into thermal energy, that is, produce photothermal effect. At the same time, local high heat also promotes the production of reactive oxygen species in the surrounding environment to cause cell apoptosis. Therefore, they can improve the therapeutic effect through synergistic treatment with photothermal therapy and photodynamic therapy. Zhao et al. synthesized carbon dots (CDs) of coronene derivatives for PTT of cancer (Figure 3b) [58]. The as-prepared CDs by virtue of the presence of large number of continuous energy bands and the narrow band gap exhibited 54.7% photothermal conversion efficiency at 808 nm. Because of excellent cell penetration capability, biocompatibility, and photostability, the CDs exhibit desirable potential for efficient PTT against cancer. Geng and co-workers proposed a zero dimension/two dimensions (0D/2D) based on carbon dot and WS_2_ [59]. The as-prepared composite (denoted as CD/WS_2_ HJs) showed desirable PCE against tumor in the second NIR window (1000–1350 nm) and synergistically enhanced optical absorption. Moreover, CD/WS_2_ HJs showed commendable deep-tissue photothermal effect to ablate osteosarcoma. Ultimately, the composite was applied in osteosarcoma treatment and bone-tissue engineering.

**Figure 3 molecules-28-00277-f003:**
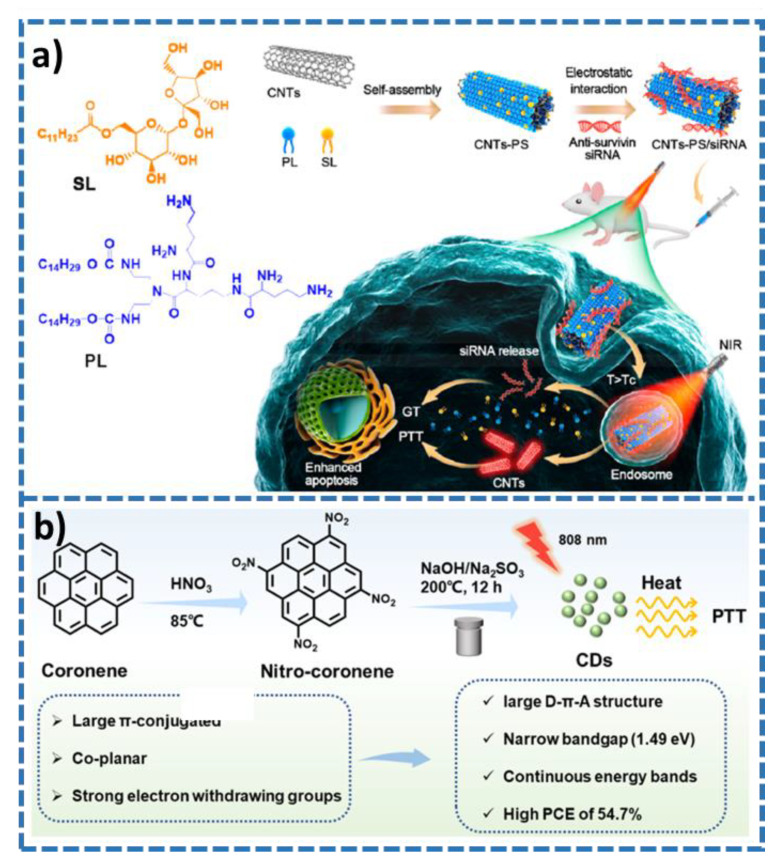
(**a**) Schematic illustration of the preparation of CNT-PS/siRNA composite and application (from Ref. [53], with the kind permission of American Chemical Society). (**b**) Synthetic route of the CDs and their application for PTT (from Ref. [58], with the kind permission of American Chemical Society).

### 2.2. Noble Metal Nanostructures

The noble metal nanostructures used in tumor photothermal therapy mainly include gold, silver, palladium, and platinum [60]. These noble metal nanostructures all have strong localized surface plasmon resonance (LSPR), which makes them have strong absorption capacity for near-infrared light and can convert the absorbed light energy into heat energy. Therefore, these noble metal nanostructures have high photothermal conversion efficiency.

#### 2.2.1. Gold

In recent years, researchers of noble metal photothermal agents have paid more attention to some nanomaterials based on gold [61,62,63], such as gold nanostars, gold nanorods, gold nanocages, gold nanoshells, gold nanospheres, etc. Gold nanoparticles used as photothermal agents have some characteristics, such as adjustable physiochemical and biochemical properties, good biocompatibility, poor biodegradability, and light stability. Based on the size, shape, and aggregation state of gold nanoparticles, as well as the modification of different biomolecules, the therapeutic efficiency of gold nanoparticles varies. The photothermal conversion effect of gold is related to the size and morphology of gold. Therefore, the required photothermal conversion materials with specific absorption wavelengths can be prepared by adjusting the size and morphology of gold. Yim et al. synthesized polydopamine-modified gold nanorods for enhanced photothermal therapy and photoacoustic imaging [64]. Dopamine-modified ultrasmall gold nanorods (SGNR@PDA) and conventional large gold nanorods (LGNRs@PDA) were synthesized. SGNR@PDA exhibited ablation of 95% of human ovarian adenocarcinoma ovarian cancer cells in vitro. As shown in Figure 4, a nanoplatform based on matrix metalloproteinase (MMP)-responsive AuNPs was synthesized by modifying complementary DNA strands onto AuNPs, tethering doxorubicin and coating poly(ethylene glycol) and a MMP-cleavable peptide, respectively [65]. Finally, the formed nanoplatform was used for enhanced photoacoustic imaging-guided PTT of tumor.

#### 2.2.2. Silver

Silver shows important application value in cancer treatment due to their low toxicity and easy preparation. The plasmon resonance of silver can be finely tuned to the near-red outer domain by developing anisotropic silver nanoparticles. Silver nanoparticles commonly used in photothermal therapy such as silver nanotriangles, silver nanospheres, and silver nanocages [66,67,68]. Rivas Aiello et al. synthesized poly(vinylpyrrolidone)-coated silver nanoplates (PVPAgNP) for PTT against cancer in HeLa cells [69]. They demonstrated cell killing by illuminating PVPAgNP loaded cells with two different photodestruction modes of continuous wave He-Ne laser and femtosecond-pulsed near infrared light, which showed great potential in PTT against cancer. A DNA/silver nanoclusters (AgNCs) was proposed for fluorescence guided PTT against tumor [70]. A theranostic platform was proposed for label-free imaging of cell surface glycans by combination of DNA/silver nanoclusters (DNA/AgNCs) and hybridization chain reaction (HCR). The as-prepared nanomaterials can absorb light and effectively convert light energy into local heat, resulting in good PTT efficacy in Figure 5. In addition, the resulting nanomaterials can be used to detect the cellular glycans and effectively kill cancer cells under imaging guide, which shows great promise for PTT-based cancer treatment.

#### 2.2.3. Other Noble Metals

Other noble metals, such as platinum and palladium, are also used in photothermal therapy in the anti-tumor field [71]. Phytic acid (PA)-capped platinum nanoparticles were reported with high affinity to hydroxyapatite and both the inherent bone-targeting ability and PTT of anticancer [72]. The use of platinum nanoparticles in PTT has the advantages of a mild hyperthermia effect and ideal chemical and thermal stability. The PA-capped platinum nanoparticles showed high affinity to hydroxyapatite in vitro and in vivo and maintained both the inherent anticancer ability of PA and photothermal effect of platinum nanoparticles. Under near-infrared irradiation, the resulting nanomaterials can effectively inhibit the growth of bone tumor and tumor related osteolysis, which provides an effective strategy for the treatment of bone malignant tumors. Palladium has the advantages of good photothermal stability, excellent hyperthermia efficiency and low cost, which makes it an ideal candidate material for PTT. Ming et al. synthesized palladium nanosheets and immunoadjuvant cytosine-phosphate-guanine oligodeoxynucleotides nanocomposites (Pd-CpG) for cancer photothermal combined immunotherapy [73]. Pd nanosheet can enhance the immunostimulatory activity and the uptake of CpG by immune cells.

### 2.3. Transition Metal Sulfides

Transition metal sulfides have attracted more attention due to their low cost and high photothermal conversion efficiency in cancer photothermal therapy [74]. Transition metal sulfides commonly used as PTAs include copper sulfide, molybdenum sulfide, and tungsten sulfide [75,76,77]. Cai et al. synthesized CuS NPs loaded with ataxia telangiectasia mutated (ATM) inhibitor and modified anti-TGF-β antibody (denoted as CuS-ATMi@TGF-βNPs) [78]. As shown in Figure 6, the as-prepared composite by virtue of synergistic chemotherapy and low-temperature PTT can effectively inhibit tumor growth. A molybdenum disulfide-based composite with copper sulfide, polyethylene glycol (PEG), and doxorubicin (DOX) as loading agents were synthesized [79]. The as-prepared material showed NIR-induced drug release behavior and good photothermal conversion efficiency, which reveal a good prospect for synergistic chemotherapy and PTT of tumor. Wang and co-workers reported PEGylated tungsten disulfide nanoparticles (WS_2_-PEG NPs) for computed tomography (CT) imaging and PTT against cancer [80]. The as-prepared nanoparticles showed high photothermal conversion efficiency and favorable photothermal ablation ability against tumor. The nanoparticles can be used as excellent contrast agents for CT imaging of tumors because of the significant X-ray attenuation of W atom. 

### 2.4. Organic Polymer

Due to the presence of delocalized π electrons, organic polymer can transfer energy and efficiently convert the absorbed infrared light energy into heat energy. Therefore, some organic materials can be used as photothermal preparations. Organic polymer employed as photothermal agents are mainly some polymer nanoparticles, such as polypyrrole, polyaniline, polydopamine, and so on [81,82,83]. Zeng and co-workers synthesized polypyrrole-based nanoparticles (PPy-PEG NPs) for PTT under the second near-infrared light (NIR II) [84]. The nanoparticles show photoacoustic/fluorescence/NIR II multimodal imaging, excellent photothermal conversion efficiency, and commendable photostability. Further, this nanomaterial displays a large amount of tumor accumulation and has a good photothermal effect, which can effectively eliminate tumors. Tian and colleagues reported a bovine serum albumin (BSA)/polyaniline (PANI) theranostic agents with pH-responsiveness toward tumor for photoacoustic imaging and PTT [85]. The nanocomposite showed amplified photoacoustic signals and satisfactory photothermal effects in tumor acidic microenvironment. As shown in Figure 7a, polydopamine-polyethyleneimine nanoparticles with a size of about 236 nm (LPPNPs) and a size of 13 nm (SPPNPs) were used to load genes to form two complexes named as LPPNPs/gene and SPPNPs/gene, respectively. LPPNPs/gene and SPPNPs/gene show gene transfection efficiency and low toxicity [86]. LPPNPs have been used to deliver the tumor suppressor gene p53 DNA for tumor therapy because of their better gene transfection ability than SPPNPs. LPPNPs proved to be a better potential synergistic photothermal therapy and gene therapy agents for the cancer treatment. In summary, organic polymer is one of the popular nanostructures used as PTAs. In comparison with other PTAs based on conventional nanostructures, organic polymer has some unique characteristics, such as low preparation cost, high optical stability, easy scale regulation, unknown distribution, and metabolic pathways.

### 2.5. Other Nanostructures

In the recent years, a few of other nanostructures/nanomaterials have been reported to serve as photothermal agents for the photothermal cancer treatment. These nanostructures include polyoxometalates (POMs), transition metal oxide (molybdenum oxide, iron oxide, tungsten oxide), and so on [88,89]. Polyoxometalates, as negatively charged and adjustable metal nanoclusters, have the advantages of precise chemical composition, clear structure, uniform size control and biocompatibility [90,91]. Shi and colleagues produced Fe-doped polyoxometalates (Fe-POM) cluster by a simple method for photothermal-enhanced chemodynamic therapy under NIR II laser (1060 nm) irradiation [92]. The resulting nanomaterials presented a strong absorption in NIR II window. Additionally, under the irradiation of 1060 nm laser, the tumor can be effectively destroyed without causing obvious damage to normal tissue. MoO_2_ nanoparticles were prepared by a hydrothermal reaction for PTT against tumor in Figure 7b [87]. The as-prepared nanomaterials exhibit excellent PTT effect. In vivo experiments showed that the administration of MoO_2_ nanoparticles in mouse neck tumors increased the tumor temperature to 66.3 °C, thereby reducing the tumor volume from ≈165 mm^3^ to ≈65 mm^3^. The nanomaterials have a significant inhibitory effect on cancer cell activity, which demonstrate its potential as a PTT nanoagent in cancer.

## 3. Conclusions and Perspective

Currently, a wide variety of nanostructures with great prospects have been introduced into photothermal tumor treatment inspired by the demand of selective and noninvasive cancer treatment. Nanostructures with different morphologies and sizes usually have different therapeutic effects against tumor. Therefore, the design and preparation of functionalized nanostructures with good therapeutic effects have been the focus of attention. Different nanostructures as PTAs have unique advantages and disadvantages, thus it is of great value to prepare the composite material with advantages of various nanostructures for photothermal therapy in tumor treatment. The recent advancement of nanomaterials/nanostructures including carbon nanostructures, noble metal nanostructures, transition metal sulfides, organic polymer, and other nanostructures as photothermal agent for photothermal therapy were discussed in the review, providing research guidance or direction for the development of novel nanostructures with various unique properties in tumor therapy.

However, the biological properties of various nanostructures as photothermal agents are still poorly understood, especially in terms of toxicity and fate after injection. These nanostructures will not appear in clinical applications until they have been shown to clear after treatment without adverse effects. The purpose of the review is to describe recent developments for photothermal therapy in tumor treatment and to provide the new strategies in the development of photothermal agents for tumor therapy. In the future, new functionalized nanostructures or more kinds of nanostructures with the ability to enhance tumor therapy will be used as photothermal agents.

## Figures and Tables

**Figure 1 molecules-28-00277-f001:**
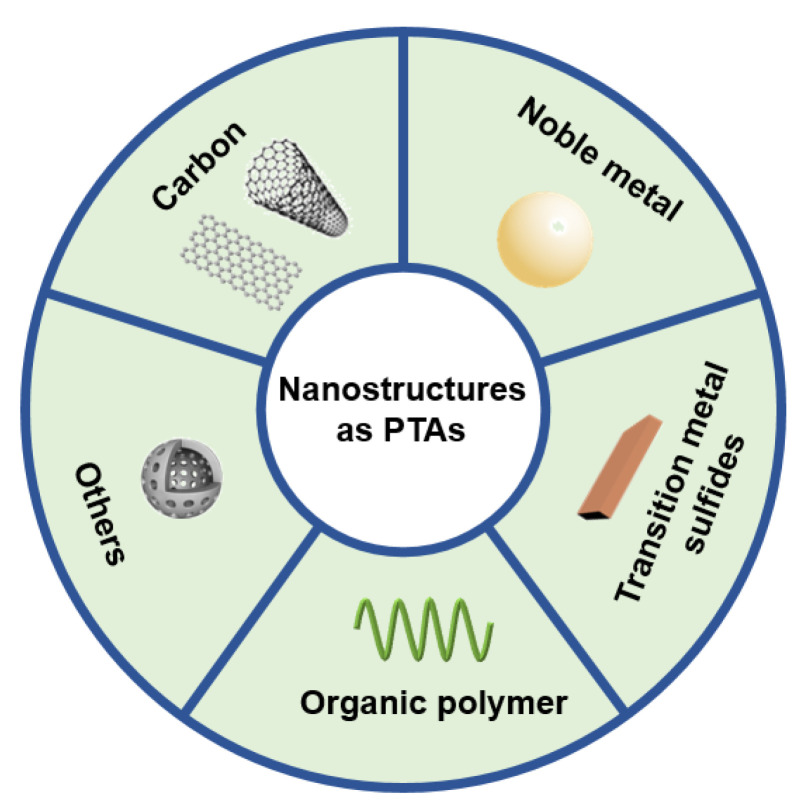
Nanostructures as photothermal agents in the anti-tumor field.

**Figure 2 molecules-28-00277-f002:**
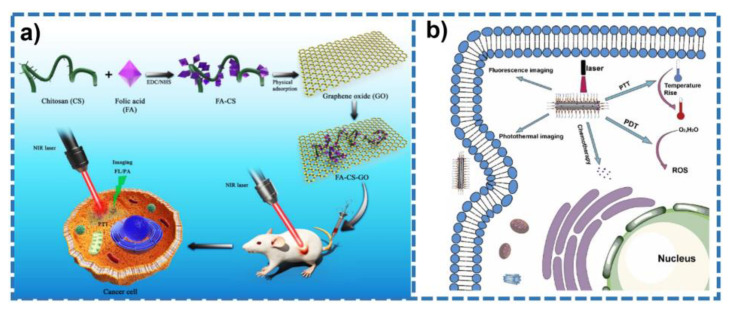
(**a**) Schematic preparation process of FA-CS-GO and its application (from Ref. [47], with the kind permission of Elsevier Ltd.). (**b**) Schematic illustration of the application of MGBP in tumor treatment (from Ref. [49], with the kind permission of Elsevier Ltd.).

**Figure 4 molecules-28-00277-f004:**
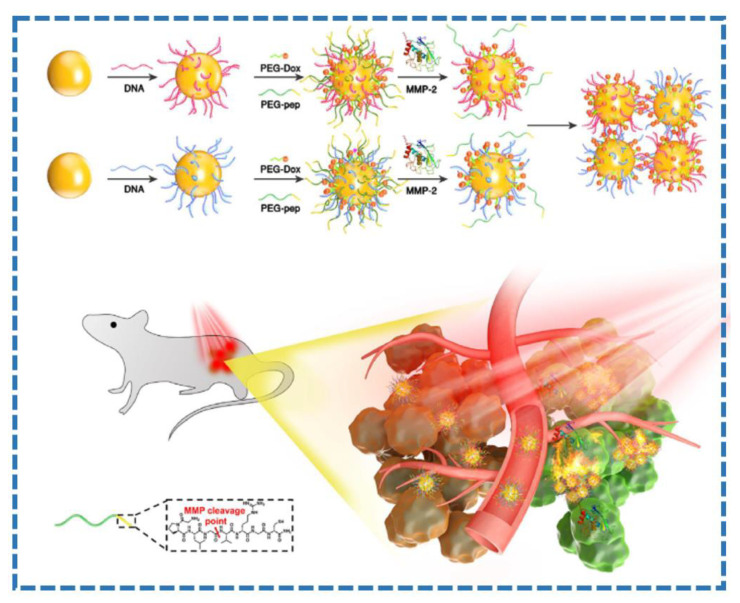
Schematic diagram of the MMP-responsive AuNPs preparation and their application for PTT against tumor (from Ref. [65], with the kind permission of Elsevier Ltd.).

**Figure 5 molecules-28-00277-f005:**
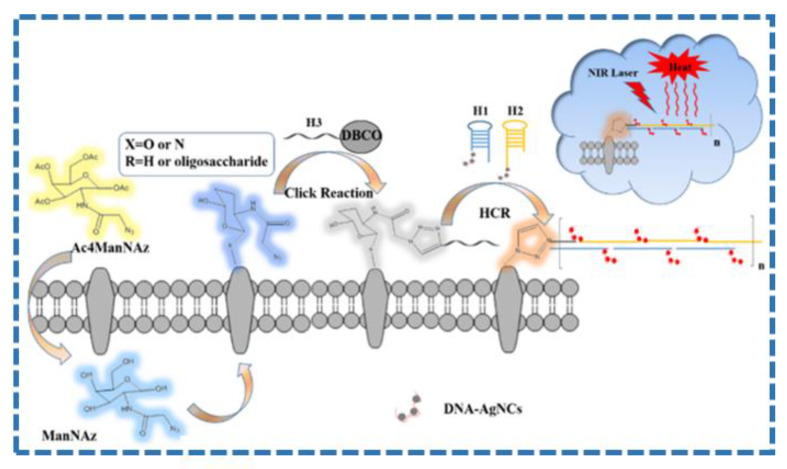
Schematic illustration of DNA/AgNCs preparation and application (From Ref. [70], with the kind permission of American Chemical Society).

**Figure 6 molecules-28-00277-f006:**
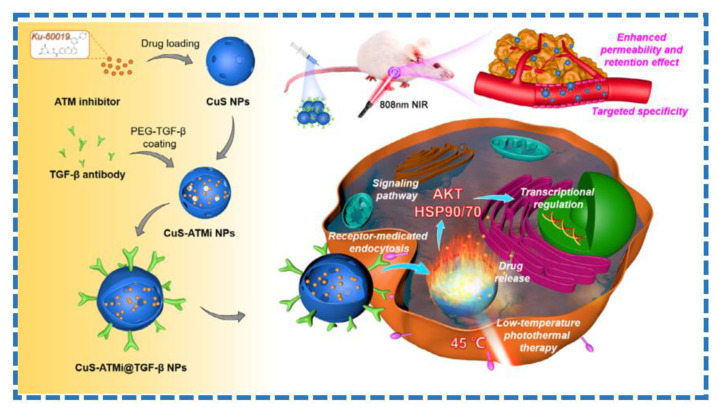
Schematic preparation of CuS-ATMi@TGF-βNPs and their application for PTT (from Ref. [78], with the kind permission of Elsevier Ltd.).

**Figure 7 molecules-28-00277-f007:**
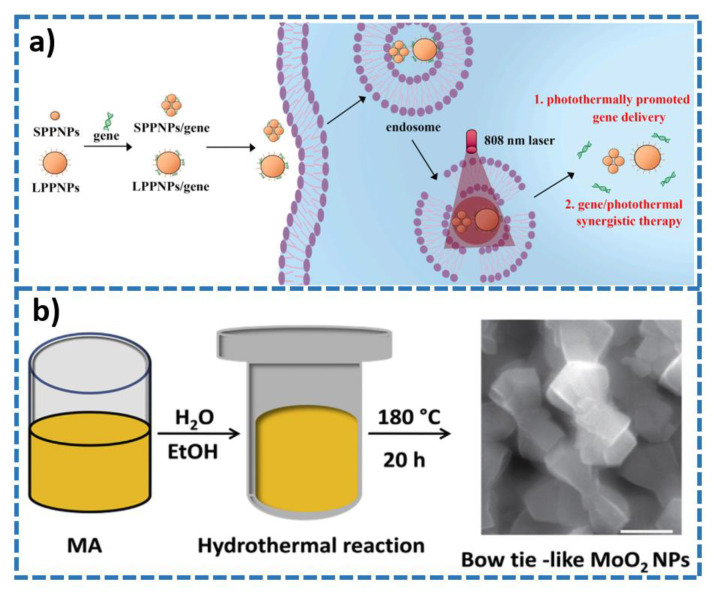
(**a**) Schematic illustration of the preparation and application of LPPNPs/gene and SPPNPs/gene (from Ref. [86], with the kind permission of Elsevier Ltd.). (**b**) The fabrication scheme of MoO_2_ nanoparticles (from Ref. [87], with the kind permission of Elsevier Ltd.).

## Data Availability

Not applicable.

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
