# Peer review of "Nanostructures as Photothermal Agents in Tumor Treatment"

_molecules, 2022, doi:10.3390/molecules28010277_

Round 1

Reviewer 1 Report

This review depicted the development of exploiting nanostructures as photothermal agents for the tumor treatment. The recently reported nanostructures and the future perspectives in the related fields are discussed. Overall, it can be accepted after some revisions.

1. The Keywords should be no less than five keywords.

2. Some abbreviations, e.g., photodynamic therapy (PDT), reactive oxygen species (ROS), etc. should also be included in the Abbreviations section.

3. The words of “photodynamic (PDT)” should be photodynamic therapy (PDT) in the manuscript.

4. Some symbols of “-”and “–” should be unified in the manuscript.

5. The first letter of some words, e.g., “Learned”, “Wrapped” in the References should be lowercase.

Reviewer 2 Report

Q1. In this paper, the content of the five groups introduced in the second part is quite sufficient. However, when giving examples in each part, the advantages of photothermal nanomaterials in tumor therapy are only mentioned. Should the disadvantages be included in the article?

Q2. After showing some information, necessary comments and reasoning judgment can be made in order to attract readers' attention.

Q3. Whether the conclusion and outlook can have more of their own understanding and views ?

Reviewer 3 Report

In this paper, the latest research progress of nanomaterials/nanostructures including carbon nanostructures, noble metal nanostructures, transition metal sulfides, organic polymer and other nanostructures as photothermal agent for photothermal therapy were summarized. It provides research guidance or direction for the development of novel nanostructures with various unique properties in tumor therapy. It is a topic of interest to the researchers in the tumor therapy but the paper still needs minor revision before acceptance for publication. My detailed comments are as follows:

1.     There are the same order number of 2.1.1 Graphene and 2.1.1 Carbon nanotubes, please check it.

2.     Please define the abbreviation of “photothermal therapy” in the place where it first appears. 

3.     In line 110 of page 3, PDT was defined repeatedly, please check it.
